# The Importance of Intra-Islet Communication in the Function and Plasticity of the Islets of Langerhans during Health and Diabetes

**DOI:** 10.3390/ijms25074070

**Published:** 2024-04-06

**Authors:** Thomas G. Hill, David J. Hill

**Affiliations:** 1Oxford Centre for Diabetes, Endocrinology, and Metabolism, Radcliffe Department of Medicine, University of Oxford, Oxford OX3 9DU, UK; 2Lawson Health Research Institute, St. Joseph’s Health Care, London, ON N6A 4V2, Canada; david.hill@lawsonresearch.com; 3Departments of Medicine, Physiology and Pharmacology, Western University, London, ON N6A 3K7, Canada

**Keywords:** islets of Langerhans, beta-cell, paracrine, plasticity, type 1 diabetes, type 2 diabetes, gestational diabetes mellitus

## Abstract

Islets of Langerhans are anatomically dispersed within the pancreas and exhibit regulatory coordination between islets in response to nutritional and inflammatory stimuli. However, within individual islets, there is also multi-faceted coordination of function between individual beta-cells, and between beta-cells and other endocrine and vascular cell types. This is mediated partly through circulatory feedback of the major secreted hormones, insulin and glucagon, but also by autocrine and paracrine actions within the islet by a range of other secreted products, including somatostatin, urocortin 3, serotonin, glucagon-like peptide-1, acetylcholine, and ghrelin. Their availability can be modulated within the islet by pericyte-mediated regulation of microvascular blood flow. Within the islet, both endocrine progenitor cells and the ability of endocrine cells to trans-differentiate between phenotypes can alter endocrine cell mass to adapt to changed metabolic circumstances, regulated by the within-islet trophic environment. Optimal islet function is precariously balanced due to the high metabolic rate required by beta-cells to synthesize and secrete insulin, and they are susceptible to oxidative and endoplasmic reticular stress in the face of high metabolic demand. Resulting changes in paracrine dynamics within the islets can contribute to the emergence of Types 1, 2 and gestational diabetes.

## 1. Introduction

The islets of Langerhans in mammalian species are dispersed throughout the pancreatic acinar tissue with a clear separating basement membrane and have a common complement of endocrine cell types derived from shared progenitor cells during embryonic and fetal development [1]. However, the anatomical distribution of endocrine cell types within the islets, their relative abundance, and their inter-cellular connectivity vary considerably between species [2]. Additionally, considerable functional diversity exists within individual endocrine cell populations and with islet size, making the integration of glycemic control extremely complicated to appreciate. The most abundant endocrine cells within mammalian islets are insulin-secreting beta-cells, followed by glucagon-secreting alpha-cells. Minority cell populations are somatostatin-secreting delta cells, ghrelin-secreting epsilon-cells and pancreatic polypeptide-secreting cells. Connectivity between these various cell types can be through the systemic circulation, the local islet vascular bed, paracrine secretions, or direct cell-to-cell contact through gap junctions. Additionally, islets are innervated by parasympathetic neurons, allowing direct control of islet function by the central nervous system. These varied signaling mechanisms can contribute to islet cell integrated function, proliferation and survival, and phenotypic plasticity to adapt to short-term metabolic demands, physiological stress such as pregnancy, or pathological processes resulting in type 1 (T1D) or type 2 diabetes (T2D). Here we review the nature of autocrine and paracrine interactions within the islets as part of normal glycemic control and how these become dysfunctional during diabetes, contributing to challenges in clinical management. The contribution to islet function by neurotransmitters originating from the islet neural network has been excluded since these represent exogenous control from the central nervous system. A schematic of the major interactions between islet cell types is shown in Figure 1.

## 2. Anatomical Architecture and Vasculature of Islets of Langerhans

In rodents, the islets of Langerhans contain a majority of beta-cells (~70%), which appear to be predominantly located within the islet core, with a surrounding rim of alpha-cells (~20%) and delta-cells (~5–10%) and occasional peripheral epsilon- and PP-cells (<1 to 5%) [2]. However, three-dimensional microscopy has revealed that the alpha- and delta-cell rim is often not continuous [3]. This anatomical arrangement would tend to maximize beta-to beta-cell interactions but result in most beta-cells not having direct cellular contact with adjacent alpha-cells. However, delta-cells have long cytoplasmic ‘neurite-like’ projections, called filopodia. These filopodia can be >10 μm in length, enabling direct paracrine contact with anatomically distant beta- or alpha-cells within the islet [4]. Most studies have focused on larger islets of greater than 100 μm diameter. However, a wide continuum of islet sizes exists within the pancreas, with the majority being of a smaller size of 50 μm or less [5]. Clearly, islet cell interactions through paracrine regulation will be more limited in these abundant, smaller islets than in the larger, more complex islets. 

Human islets anatomically differ from those of rodents when viewed by two-dimensional microscopy, appearing to have a relatively lower proportion of beta-cells (~50%) but more alpha-cells (~30–40%), whereby the islet architecture comprises a more intermingled arrangement of alpha- and delta-cells throughout the islets, as opposed to residing in the mantle as seen in rodent islets [2,6]. However, this simplistic form of analysis may be misleading as considerable diversity exists in the relative proportion of beta-cells within individual human islets, with a higher proportion of alpha-cells being characteristic of larger islets only [7,8]. There is also evidence that in larger human islets, sub-islet anatomical clusters exist, comprising a core of beta-cells surrounded by alpha- and delta-cells [9]. The alpha-cells were found to be equally distributed between the islet periphery and adjacent to the islet core capillaries, but PP-cell presence was higher in the islet mantle while delta-cells were more abundant in the core [10]. No differences were found between the head and tail of the human pancreas with respect to delta-cell presence, although PP cells are more abundant in the tail. A surprising recent observation was that, using three-dimensional imaging, 50% of insulin-expressing islets had little or no alpha-cells, suggesting that intra-islet communication between alpha- and beta-cells is likely to show enormous heterogeneity between islets [11]. As in rodent islets, human delta-cells also exhibit long filopodia, allowing them to interface with extensive populations of beta-, alpha-, and other delta-cells, as well as vascular endothelial cells [4]. A report by Arrojo E Drigo et al., [4] also provided evidence that the length of the delta-cell filopodia was regulated by locally secreted insulin-like growth factor-1 (IGF-1) and vascular endothelial growth factor (VEGF).

Although the islets represent only around 2% of pancreatic cellular mass, they receive ~20% of the arterial blood flow through a closely packed network of branching capillaries [12], although vascular density is greater in mouse islets than in humans [13]. High vascularity allows the majority of islet endocrine cells to have close contact with a fenestrated endothelial cell layer for their rapid ability to sense prevailing glucose concentrations and to export secreted peptides [14]. Beta-cells are polarized with a basal domain adjacent to the capillary, where most insulin granule exocytosis occurs, and an apical domain away from the capillary [15]. They do not make direct contact with endothelial cells since a basement membrane surrounds the islet capillaries. The islet vasculature consists of small arterioles that penetrate the islet rim and form a capillary bed within the islet core, with venous flow exiting from the core to the mantle [16]. While recent reports have found considerable diversity between individual islets in blood flow directionality, the predominant core-to-mantle arrangement would favour insulin and other beta-cell secretory products to be transported to the alpha- and delta-cells rather than the reverse [16]. Islet blood flow can therefore act as an independent regulator of islet cell function and plasticity. It can be differentially regulated within individual islets or regions within an islet through the actions of contractile pericytes embedded within the basement membrane surrounding the capillaries in response to glucose levels, or neural control through epinephrine release [17,18]. Targeted depletion of pericytes in mice through the expression of diphtheria toxin has been shown to cause a reduction in beta-cell insulin content and secretion, reduce the expression of maturity-associated beta-cell transcription factors such as MafA and Pdx1, and lead to glucose intolerance [19].

## 3. Functional Heterogeneity of Islet Endocrine Cells

Considerable biochemical coordination exists between beta-cells to orchestrate glucose-stimulated insulin secretion (GSIS). This is mediated by rapid oscillations in intracellular Ca^2+^ that can spread across the beta-cell population of an individual islet due to direct connectivity through Connexin-36 (Cx36) gap junctions [20]. Beta-cell pairs secrete five times more insulin than single beta-cells, and beta-cells within islets secrete around 18-fold more insulin per cell than physically isolated cells [21,22]. ‘Hub’ or ‘first responder’ beta-cells represent 1–10% of the mouse islet beta-cells and are initial responders to trophic stimuli of GSIS and orchestrate the inter-cellular wave of intracellular Ca^2+^ [23,24]. Compared to other beta-cells, hub cells have enhanced levels of key metabolic hormones such as glucokinase (GCK), less stored insulin and are phenotypically more immature, as defined by the expression of key transcription factors such as Pdx1. A similar heterogeneity is predicted to also exist in human islets [25]. Even among non-hub beta-cells, considerable variability can exist between individual cells in terms of the expression of genes related to insulin expression and metabolic release, such as PDX1, GCK, and glucose transporter-2 (GLUT2), in both mice and humans [26,27], which may relate to their contribution to total GSIS or the glucose threshold of activation. More than 50% of beta-cells in human islets that are highly glucose responsive demonstrated a higher expression of CD9, a gene encoding a cell surface tetraspanin family transmembrane glycoprotein associated with cell adhesion and differentiation, and ST8SIA1, encoding a subtype of the integrin alpha 1 gene also associated with cell–cell adhesion, while lower expression levels of these genes characterized around 10% of cells that were less responsive to glucose [28]. 

Similarly, identified sub-populations of beta-cells may have distinctly different potential with respect to proliferation or survival. The gene Flattop (FLTP) is a modulator of islet endocrine cell polarity and contributes to epithelial architecture [29]. FLTP-positive beta-cells increase as a percentage of beta-cell mass in mice with advancing age until adulthood, where they represent 80% of cells. Their eventual phenotype has a high expression of genes associated with mature GSIS potential, such as urocortin-3 (UCN3) and the transcription factor MAFA [30]. FLTP-negative beta-cells have a greater proliferative potential and can increase in number under metabolic stress as an adaptive mechanism to increase beta-cell mass, such as during pregnancy. The relative presence of high or low expression of Glut2 in mouse islets can also discriminate between mature and immature, but more proliferative beta-cell populations. An islet ‘niche’ of immature beta-cells has been suggested to exist in the islet mantle area in rodents but is dispersed within the islets in humans [31]. These suggested progenitors are characterized by a relatively low expression of insulin and Glut2 but a higher rate of proliferation in vivo [32]. Their abundance decreases with age. We previously identified that these progenitors contributed to the recovery of beta-cell mass in young, streptozotocin (STZ)-induced diabetic mice, since they were not targeted for destruction by STZ, and to the expansion of beta-cell mass during mouse pregnancy [33,34]. Whether Glut2-low cells can transform into Glut2-high, mature beta-cells is not yet clear. When Glut2-low cells were harvested from the neonatal mouse pancreas, we found that such maturation could occur in vitro [32], but Glut2-low cells did not transform into Glut2-high cells following STZ treatment in adult, diabetic mice [35].

Heterogeneity also exists within non-beta-cell islet cell populations. Single-cell transcriptomic analysis has shown a number of sub-clusters of alpha-cells with widely differing transcription factor and preproglucagon gene expression [36], and at least two populations of human alpha-cells were reported with different glucagon content [37]. Additionally, sub-populations were identified with a differing proliferative-associated gene expression phenotype [38]. A population of both mouse and human delta-cells also co-express the hormone PPY [39]. These bi-hormonal cells did not appear to contribute to glucose homeostasis, but in diabetic mice they gained the ability to express insulin. Interestingly, discrete juxta-positioned pairs of delta- and beta-cells exist where the beta-cells have a distinct molecular transcriptome. This may enable such beta-cells to preferentially re-enter cell proliferation following an experimental reduction in beta-cell mass.

## 4. Paracrine Islet Cell Interactions in Normal Physiology

### 4.1. Beta-Cells

Insulin and co-secreted molecules from the beta-cells have major paracrine effects on the secretion of other endocrine-derived hormones. Insulin itself has direct actions on adjacent alpha-cells to reduce glucagon secretion [40]. If, as suggested by studies of islet microvasculature, many islets have a directional blood flow from the core of the islet to the mantle, and given that the local concentration of insulin within the islets is in the high nanomolar range, then the paracrine actions of insulin may influence much of the islet. Pancreatic perfusion with insulin secretagogues was shown to prevent the release of glucagon in response to hypoglycemia in humans [41]. The binding of insulin to insulin receptors (INSRs) on alpha-cells results in activation of ATP-regulated K^+^ (K_ATP_) channels, hyperpolarization of the cell membrane and a decrease in intracellular Ca^2+^, resulting in less glucagon release [42]. The actions of insulin on glucagon release depend on insulin being bound to zinc [43], although application of zinc alone did not have any effect on glucagon release from mouse alpha-cells [44], and mice null for the zinc transporter, Slc30a8, showed normal glucagon release kinetics [45]. Insulin may also have autocrine actions on beta-cells as a positive feedback regulator to increase insulin gene transcription through activation of the insulin receptor substrate 2 (IRS2) and the phosphoinositol-3 kinase (PI3K) signaling pathway [46]. It is unclear if these actions are mediated through the insulin or insulin-like growth factor-1 (IGF-1) receptors. While insulin receptors are present on beta-cells, the high local concentrations of insulin in adjacent extracellular fluid would be capable of binding to and activating IGF type 1 receptors. Insulin has been reported to promote glucose-dependent cell proliferation in MIN6 mouse beta-cells and to prevent apoptosis [47]. However, exogenous IGF-2 has similar trophic actions on beta-cells [48]. Beta-cell-specific insulin receptor knock-out (BIRKO) mice showed reduced GSIS, glucose intolerance and a lower beta-cell mass, showing that at least some of the metabolic and trophic actions of autocrine-acting insulin must be mediated through the insulin receptor [49,50]. 

Beta-cells directly connect with the filopodia of delta-cells through gap junctions, and beta-cell secretion products can stimulate somatostatin release, resulting in the paracrine inhibition of insulin and glucagon release. However, this may not be primarily due to the direct actions of insulin but mediated through UCN3, which is co-secreted in the insulin granules in a glucose-dependent manner [51]. UCN3 is related to corticotrophin-releasing hormone (CRH) and activates the CRH type 2 alpha receptor (CRHR2a) that is selectively expressed on delta-cells [51]. In rodents, Ucn3 expression is restricted to beta-cells within the islets, but in humans, it is expressed in both beta- and alpha-cells and at lower levels by PP cells [52,53]. Exogenous UCN3 increases somatostatin release from delta-cells and a specific CRHR2a antagonist reduces glucose-stimulated somatostatin release from perifused islets [51,54]. Ucn3 null mice exhibited reduced islet somatostatin expression and content, as well as a reduction in islet delta-cell number [55]. During the development of the pancreas, Ucn3 expression in mouse islets appears only at the end of gestation, and expression increases in parallel with the functional maturity of beta-cells as they become acutely glucose-sensitive [56,57].

Gamma-aminobutyric acid (GABA) is also a secretory product of the beta-cell involved in the paracrine inhibition of glucagon secretion by alpha-cells [58]. While GABA was previously thought to be released from discrete secretory vesicles by beta-cells that are distinct from insulin-containing granules, under similar glucose-stimulated secretion [59], the major secretory route may be continuous but pulsatile release from a cytoplasmic pool that is not dependent on glucose [60]. A continuous pulsatile release of GABA would suggest that a major paracrine role is to fine-tune minute-by-minute adjustments to glucagon and insulin release rather than contributing to large, episodic changes post-prandially. GABA activates GABA_A_ receptors on alpha-cells, which are ligand-gated Cl^−^ channels that hyperpolarise the alpha-cell and reduce glucagon secretion [61]. 

Beta-cells also express tryptophan hydroxylase and can synthesize the neurotransmitter serotonin (5-hydroxytryptamine—5-HT), which is packaged into insulin granules and co-released under glucose control [62]. 5-HT can act as a paracrine agent to inhibit glucagon release from alpha-cells by binding to 5-HT_1F_ receptors and signaling via G_alpha i/o_ G protein to lower intracellular adenylyl cyclase activity and cAMP production [18]. It can also promote beta-cell proliferation in early life and during the beta-cell expansion seen in the mother during pregnancy [63].

### 4.2. Alpha-Cells

When blood glucose becomes too low (below ~4 mM), systemic glucagon secretion from alpha-cells has a major effect on elevating plasma glucose levels through mobilizing glucose production from the liver by gluconeogenesis and glycogenolysis, which in turn gives rise to an increase in GSIS from the beta-cells. Conversely, islet glucagon secretion is inhibited by elevated glucose, whereby the concentration-dependent suppression of glucagon is maximal at 6 mM glucose in isolated islets [64]. The mechanisms underlying the regulation of glucagon secretion remain debated but are likely to involve: positive regulation by non-islet-derived paracrine factors, such as glucose-dependent insulinotropic polypeptide (GIP), intra-islet neurotransmitters acetylcholine (via muscarinic receptors), and adrenaline (via beta-adrenergic receptors); positive regulation by amino acids, such as alanine and arginine (via electrogenic entry) and non-esterified fatty acids; as well as an intrinsic regulation via glucose metabolism [65]. Under mild hyperglycemic conditions experienced post-prandially, glucagon has direct paracrine actions on the beta-cell to stimulate insulin release [66]. This action is largely mediated through both glucagon and glucagon-like peptide-1 (GLP-1) receptors on beta-cells and the subsequent activation of the G_αs_ G protein subunit, resulting in the generation of cyclic AMP and an influx of cellular Ca^2+^ leading to insulin granule secretion [67,68]. Under hypoglycemic conditions, glucagon is unable to trigger GSIS due to the dependency of this action on a threshold level of glucose presence [69]. Targeted genetic deletion of the glucagon or GLP-1 receptors from beta-cells results in impaired GSIS and glucose intolerance, while over-expression of the glucagon receptor causes enhanced GSIS, demonstrating this paracrine signaling pathway to be physiologically important [67,70,71]. 

Paracrine glucagon signaling from alpha-cells is most likely to occur in alpha-cells directly juxtaposed to beta-cells within islets across the intervening extracellular fluid, since the prevailing vascular flow, at least in rodents, is considered to be from the beta-cell-rich core to the mantle, where most alpha-cells reside. However, in the context of human islets, alpha- to beta-cell intercellular signaling may be more widespread given the existence of regional alpha-/beta-cell foci within larger islets and the wide variation in net direction of vascular flow being reported between the core and mantle for individual islets. In contrast, the finding that up to 50% of smaller human islets have fewer alpha-cells would indicate that the indirect systemic route by which glucagon leads to increased GSIS must be dominant for at least this islet population [11]. 

If GLP-1 is of physiological relevance in contributing to the paracrine stimulation of GSIS postprandially, this is unlikely to mostly derive from the intestinal L-cells, since circulating GLP-1 has a short biological half-life of a couple of minutes as a result of degradation by dipeptidyl peptidase 4 (DPP-4) [72]. It seems more likely that GLP-1 derives from the alpha-cells as an alternatively spliced product of preproglucagon due to the actions of the prohormone convertase 1/3 (PC1/3). While glucagon is the major secreted protein following splicing of preproglucagon with prohormone convertase 2 (PC2), GLP-1 is produced in smaller amounts by PC1/3 [73,74], although this pathway may be preferred as an adaptive mechanism to increase beta-cell proliferation and/or survival during metabolic stress or following beta-cell loss [75]. In a mouse model whereby PC1/3 was selectively deleted in alpha-cells, islet GLP-1 content was reduced by a third and associated with impaired glucose tolerance when the animals were challenged with a high-fat diet [76]. While glucagon can stimulate GSIS at the beta-cell through either the glucagon or the GLP-1 receptor, the glucagon receptor is the main mediator of glucagon actions on hepatic metabolism [67]. The GLP-1Rs are also expressed by the alpha- and delta-cells, whereby GLP-1-GLP-1R signaling has been shown to inhibit alpha-cell glucagon secretion both directly (by alpha-cell GLP-1-GLP-1R-induced inhibition of alpha-cell P/Q-type Ca^2+^ channels) [77] as well as indirectly via delta-cell GLP-1-GLP-1R-induced stimulation of somatostatin release [78]. The pancreatic islets also express DPP-4, suggesting the intra-islet breakdown of alpha-cell-derived GLP-1(7–36) into GLP-1(9–36). Unlike GLP-1(7–36), GLP-1(9–36) does not potentiate GSIS from beta-cells. Although previously assumed to lack biological activity, a recent report has shown that GLP-1(9–36) also exerts a strong glucagonostatic effect on alpha-cells, although exclusively by activating alpha-cell glucagon receptors (GCGRs) [79].

Other trophic molecules released by the alpha-cells include acetylcholine [80] and CRH [81]. As with acetylcholine release from parasympathetic innervation of the islets, alpha-cell-derived acetylcholine has been shown to enhance GSIS at the beta-cell [80]. Beta-cells express CRH type 1 muscarinic receptors, which can activate the G_αs_ G-protein subunit and can, at least experimentally, contribute to GSIS [82]. Acetylcholine is released during the cephalic phase after feeding from parasympathetic nerves innervating the islets, prior to an increase in blood glucose [83], but it is not known if this also applies to alpha-cell-derived acetylcholine. However, the cholinergic innervation of human islets is considerably less than in rodent islets [84], and postprandial serum insulin levels are unaltered in vagotomised patients [85], suggesting that alpha-cell-derived acetylcholine may be a physiologically relevant contributor to glycemic control. However, paracrine signaling to beta-cells by acetylcholine must be transient, as it is rapidly degraded in the extracellular fluid by cholinesterases. In addition to the beta-cells, intra-islet acetylcholine has also been shown to stimulate somatostatin secretion from the delta-cells, contributing to a negative feedback inhibition on beta-cell insulin release [86]. 

Glucagon receptors are also expressed by delta-cells, and glucagon has been shown to increase somatostatin release from the perfused dog pancreas [87]. GCGR signalling on delta-cells is important for regulating delta-cell mass, as delta cell proliferation is observed in glucagon receptor null mice [88]. Additionally, secreted glucagon may regulate its own release through an autocrine negative feedback action on the alpha-cell, as also seen for GLP-1 action on glucagon secretion [89]. Mice lacking the glucagon receptor exhibited alpha-cell hyperplasia and hyperglucagonemia [90]. A second autocrine regulatory pathway may involve glutamate, which is co-released with glucagon in secretory granules. The alpha-cells of multiple species, but not beta-cells, express alpha-amino-3-hydroxy-5-methyl-4-isoxazole proprionate (AMPA)/kainate ionotropic glutamate receptors [91], whose activation promotes an increase in glucagon secretion [92].

### 4.3. Delta-Cells

Somatostatin released from the delta-cells is a paracrine regulatory inhibitor of both insulin and glucagon release from the beta- and alpha-cells, respectively [31,93]. Somatostatin is also released from the enteroendocrine D cells in the gastrointestinal tract as well as the central nervous system, where circulating levels range between 5 and 25 pM, which is an order of magnitude lower than is required to half-maximally activate somatostatin receptors [94]. When coupled with the observation that pancreatectomy does not alter circulating somatostatin levels [95], this suggests that somatostatin actions within the islets are predominantly due to islet delta-cell release. The major molecular form of somatostatin released in the GI tract is Sst-28, whereas the form released from delta-cells is Sst-14 [96]. 

In mouse islets, glucose-induced somatostatin secretion occurs at glucose concentrations as low as 3 mM, with half-maximal stimulation occurring at 5–6 mM [64,97]. The regulation of delta-cell somatostatin secretion includes positive modulation by intrinsic glucose metabolism and amino acid uptake (such as arginine, lysine, and leucine), and negative modulation by non-esterified fatty acids (such as palmitate) [98]. Elevations in blood glucose coordinate an increase in Ca^2+^ spiking activity across the delta-cell population that precedes an increase in somatostatin release [4]. There are five different somatostatin receptors (SSTRs1-5) expressed in rodents and humans. While SSTR2 is the most functionally dominant SSTR expressed by alpha-cells, beta-cells predominantly express SSTR3 [99]. The SSTRs are G protein-coupled receptors (GPCRs) coupled to an inhibitory G_αi_ G-protein, whose activation leads to a downstream reduction in intracellular adenylyl cyclase activity, resulting in lower intracellular cAMP, and less cAMP-induced exocytosis. Activation of the SSTRs also inhibits cell voltage-gated Ca^2+^ channel activity and activates G-protein-activated inward rectifier K^+^ channels (GIRK), which culminate in membrane repolarisation and the suppression of cell action potential firing [100,101]. In addition, somatostatin has been shown to exert a direct inhibitory effect on cell exocytosis via a mechanism independent of intracellular cAMP involving the activation of the protein phosphatase, calcineurin [102]. Since long filopodia extend from delta cells to distant alpha- and beta-cells, the impact of a glucose-dependent stimulation of somatostatin release can be both rapid and islet-wide, resulting in a decreased secretion of both glucagon and insulin. The intercellular connectivity between delta- and beta-cells is facilitated by gap junctions, predominantly involving Cx36, which allows for tight synchronicity and the electrical coupling of ions and small molecules between pulses of insulin and somatostatin [103]. 

Somatostatin inhibits glucagon release below the circulating glucose threshold (as low as 3 mM glucose) necessary to activate insulin release, which suggests an important basal action of somatostatin within the islets to limit glucagon secretion [93,104]. Paracrine interactions between the beta-, alpha- and delta-cells within the islets are considered to be responsible for determining the glycemic set point for euglycemia, and this may be led by somatostatin release [105]. In mice, the threshold for insulin release is around 7–8 mM glucose, at which concentration glucagon release is fully repressed. However, somatostatin release is active throughout a full range of glucose concentrations [93,106]. Mouse and human delta-cells also express SSTR1 and SSTR3, suggesting that somatostatin secretion itself is under autocrine inhibitory feedback, accounting for the observation of potent somatostatin secretion in the presence of somatostatin receptor antagonists [93]. Interestingly, the presence of islet delta-cells is critical for normal glycaemic control, as ablation of delta-cells in mice was shown to cause a hypersecretion of insulin, resulting in hypoglycaemia and death [107].

Outside of paracrine control of glucose homeostasis, delta-cell-derived somatostatin may also enhance beta-cell survival, as tissue culture of isolated mouse islets or MIN6 beta-cells in the presence of exogenous somatostatin reduced the presence of biochemical markers of stress and apoptosis caused by lipotoxicity, pro-inflammatory cytokines, or hypoglycemic conditions [108]. In addition, a recent study of human delta-cells identified the co-expression and presence in secretory granules with somatostatin of an alternatively spliced insulin gene product that encodes the insulin signal peptide and the B-chain [109]. The translated product was able to attract cytotoxic T lymphocytes, suggesting a contribution to islet inflammation, although the peptide was not expressed by beta-cells. The relevance of this delta-cell product to the development of T1D diabetes is unknown.

### 4.4. Epsilon and PP Cells

Ghrelin is secreted by epsilon cells within the islets, although its major site of production is the fundus of the stomach [110]. As well as stimulating the release of growth hormone from the anterior pituitary by signaling through the growth hormone secretagogue receptor (GHS-R), ghrelin also stimulates appetite by acting on the hypothalamic arcuate nucleus [111], thereby acting as a counter-hormone to leptin in nutritional control. Islet epsilon cell abundance is greatest in fetal development but is much reduced in adult humans [112]. Ghrelin can be processed in an acetylated or unacetylated form, but acetylation is necessary for binding to the ghrelin receptor R1a (GHS-R1a) [113]. This receptor is highly expressed by delta-cells and, to a lesser extent, alpha- and beta-cells in both rodent and human islets [99,114] suggesting paracrine interactions. In addition to acetylated and unacetylated ghrelin, a third peptide, obestatin, is generated by post-translational processing of the preproghrelin protein precursor [115]. Deletion of ghrelin-secreting epsilon cells in mice results in increased GSIS, while administration of exogenous ghrelin decreased insulin secretion and elevated blood glucose levels [112,116]. In ghrelin-null mice, basal insulin release was unaltered, and insulin gene expression and insulin content were not changed, suggesting that the primary action is on GSIS [117]. However, reports are inconsistent, and Gray et al. [118] found no change in amino acid or GLP-1-stimulated insulin release in either ghrelin or GHS-R adult null mice. Given the low abundance of epsilon cells in the adult pancreas, this may reflect a limited action of paracrine ghrelin on adjacent beta-cell populations only. While direct actions on insulin release can clearly occur [119], the ability of ghrelin to suppress GSIS and also glucagon release has been shown to be largely indirect through increased somatostatin release from the delta-cells [113,120]. Ghrelin may also indirectly alter GSIS by acting as a local vasoconstrictor on arterioles within the islets to alter vascular flow [121]. The GHS-1 receptor is unusual in that it has constitutive signaling activity in the absence of ghrelin ligand and may therefore contribute to the basal tone of islet blood flow and endocrine hormone secretion [122]. Constitutive signaling can be regulated through the inhibitory actions of a liver-derived GHS-1 ligand, liver-expressed antimicrobial peptide 2 (LEAP2), whose circulating levels are regulated by food intake and correlate with body weight [123]. Unlike ghrelin, obestatin has a predominant positive action on insulin release at low glucose levels but a ghrelin-like inhibitory effect at higher glucose levels [124].

In addition to fine-tuning GSIS, ghrelin can also exert trophic actions directly on the beta-cells by increasing proliferation and inhibiting apoptosis [125], both mediated through activation of the PI3K-Akt signaling pathway [126]. This would perhaps explain the major presence of epsilon cells in early life when beta-cell proliferation is maximal. In fetal islets, ghrelin appears to be released from a more immature phenotype of endocrine cell that co-expresses both glucagon and pancreatic polypeptide [127]. However, obestatin may be the most bioactive gene product within the context of islet cell proliferation and survival as it can advance beta-cell lineage development from pancreatic progenitors [128]. Such actions may be mediated, in part, by the angiogenic role of obestatin in the developing islets [129]. The trophic actions of exogenous ghrelin and obestatin are also seen on islets postnatally, as both can limit STZ-induced beta-cell death in rats [130].

Gamma-, or pancreatic polypeptide (PP), cells secrete pancreatic polypeptide, a member of the NPY family of peptides that is a ligand for the NPY receptor 4 [131,132]. NPYR4 signaling is associated with regulation of energy metabolism and satiety at the hypothalamus and control of exocrine pancreatic secretions, but the receptor is also expressed within islet delta-cells and mediates an inhibitory action of PP on somatostatin release [133]. Exogenous PP inhibited not only somatostatin release but also that of glucagon and insulin [133,134,135]. Since PP is rapidly degraded in the circulation by DPP-4 and other proteolytic enzymes, any such physiological action of PP is likely to be from paracrine sources. PP is released within a few minutes of feeding, and regulatory control is most probably neural (since vagotomy substantially reduces PP release), which is mediated through the actions of acetylcholine [136,137]. However, a paracrine relationship also appears to exist between epsilon- and gamma-cells, whereby ghrelin can negatively regulate PP secretion from rodent islets via signaling through the GHS-R receptor [138,139]. Many PP-cells are bi-hormonal, and in mice and humans can co-express glucagon, insulin, or somatostatin [39].

## 5. Endothelial Cells and Pericytes

Independent of their effects on islet blood flow through capillary vasoconstriction or vasodilation, pericytes can contribute to the paracrine regulation of GSIS. Pericytes release nerve growth factor (NGF) in a glucose-dependent manner. The beta-cells express the tropomyosin receptor kinase-A (TRKA) nerve growth factor (NGF) receptor [140]. Activation of TRKA causes insulin granule exocytosis through intracellular actin remodelling [140]. Pericytes also release bone morphogenetic protein 4 (BMP4) that has trophic actions on beta-cells through the induced expression of the transcription factor 7-like 2 (TCF7L2) gene [141]. Additionally, endothelial cells secrete connective tissue growth factor (CTGF) and thrombospondin [142]. CTGF contributes to beta-cell proliferation during the increase in islet-cell mass in early life [143], while thrombospondin release is regulated by glucose and has anti-angiogenic actions [144]. Mice null for thrombospondin exhibit pancreatic hyperplasia and impaired GSIS [145]. Hepatocyte growth factor (HGF) is secreted by endothelial cells and promotes beta-cell proliferation in vitro [146]. Reciprocally, beta-cells release vascular endothelial growth factor-A (VEGF-A), which binds to the VEGFR2 receptor on adjacent endothelial cells to maintain microvascular density [147].

## 6. Islet Paracrine Disruption in T2D

Since the diverse paracrine control of islet function in normal physiology is highly dependent on the unique architecture of islets and their vascular supply, it is not surprising that changes that occur during T2D are linked to anatomical and vascular changes as well. A summary of islet paracrine changes reported to occur during diabetes is shown in Table 1. The development of T2D, exacerbated by pre-existing obesity and/or aging, can progress from initial peripheral insulin resistance to beta-cell dysfunction and ultimately death, driven through the detrimental effects of gluco- and lipotoxicity, cellular stress, inflammatory processes, and islet amyloid deposition [148]. Analyses at autopsy have shown beta-cell mass to be reduced by up to ~60%, depending on the time before diagnosis of T2D [149]. As in the liver, the pancreas accumulates higher fat content in T2D, which can precede the onset of diabetes in animal models [150], and in a human context, may contribute to the impact of obesity as a diabetes risk factor. Quantification of intra-islet capillary density in human and mouse tissues showed capillary thickening and fragmentation, but increased vessel density in T2D relative to non-diabetic controls [13]. This occurs despite a decrease in beta-cell mass in individuals with type 2 diabetes to some 60% of control values [150,151], although this may vary considerably depending on the duration of the disease. Additionally, progression of T2D is associated with a progressive loss of pericytes around islet capillaries, which will compromise the active control of islet blood flow [152]. The increased frequency of hyperglycemia also leads to dysfunction in protein glycation in both endocrine and vascular cells and an accumulation of advanced glycation end-products (AGEs), particularly within basement membrane components around the beta-cells, leading to thickening and fibrosis [153], which could impede paracrine signalling. Direct actions of AGEs on islets may also cause detrimental effects, as beta-cells exposed to AGEs in vitro show impaired GSIS associated with increased oxidative stress [154]. Similarly, T2D is associated with an accumulation of hyaluronan within islets associated with impaired insulin release [155].

A common feature of T2D is the deposition of beta-cell-derived amyloid within the islets. Amylin is co-secreted with insulin and normally exerts diverse systemic actions, including modulation of appetite, gastric emptying, and glucagon release [156]. Under normal physiological conditions, islet amyloid polypeptide (IAPP) is soluble, but in T2D (because of beta-cell molecular stress), amylin secretion increases and misfolding occurs, causing oligomerization to form amyloid fibrils [157]. Increased beta-cell stress associated with T2D and prolonged hyperglycemia is mediated by several mechanisms, including endoplasmic reticular and oxidative stress, culminating in a loss of beta-cell function [158]. In human islets, amyloid deposition occurs in close proximity to islet capillaries, leading to vessel expression of pro-inflammatory genes, structural disruption, and apoptosis [159]. Amyloid is also cytotoxic to beta-cells, resulting in disrupted GSIS and cellular death by apoptosis [160]. However, rodent IAPP does not form fibrils in vivo, and animal models of amyloid actions in T2D must utilize animals transgenic for human IAPP [161]. Since T2D is associated with advancing age, it is also likely that beta-cell aging, exacerbated by the pro-inflammatory environment within the islets during diabetes, contributes to beta-cell failure and loss. Cellular senescence, termed senescence-associated secretory phenotype (SASP), occurs in response to oxidative stress and DNA damage, such that a higher proportion of beta-cells are senescent in islets from patients with T2D than in age-matched non-diabetics [162]. Studies using both mouse and human senescent beta-cells showed a distinct transcriptional profile associated with aging and T2D [163]. The onset of SASP in beta-cells during T2D is likely to severely disrupt intra-islet signaling between beta-cells and both alpha- and delta-cells.

The combination of the need for hypersecretion of insulin to counteract the insulin resistance of T2D, the beta-cell dysfunction resulting from gluco- and lipotoxicity, amyloid toxicity, and disrupted microvasculature causes profound changes to the pulsatile nature of insulin release and the fine paracrine communication between beta-cells and other islet cell types. Outside of diabetes, regional connectivity networks of beta-cells within human islets, led by hub cells, create integrated waves of increased intracellular Ca^2+^ through gap junctions, leading to a coordinated GSIS response. While in T2D, glucose-dependence of insulin release can be retained, Ca^2+^ wave integration across beta-cells is impeded, and local integration networks become disjointed [164], contributing to a relative loss of first-phase insulin release. While the loss of the integrating activity of hub or leader cells is likely to be a major contributor, changes within the extracellular matrix and capillary efficiency in T2D that impede cell-to-cell communication and rapid paracrine signaling through the islet capillary beds are also likely to be involved. For the beta-cell, this disrupts the local effectiveness of secreted signaling molecules such as GABA and serotonin, which in isolated human islets from subjects with T2D had a reduced islet content relative to non-diabetic subjects and showed negative correlations with the HbA1C values of the donors [165]. UCN3 secretion is reduced early in pre-diabetes, with the associated loss of UCN3-stimulated somatostatin release at the delta-cell causing the loss of an important element of rapid paracrine hormone control (such as at the alpha-cells), which will contribute to glycemic instability [51]. However, in the short-term, it may be an adaptive mechanism to enhance insulin secretion to counteract the beta-cell dysfunction of T2D. 

Type 2 diabetes is associated with increased alpha-cell secretion of glucagon, leading to inappropriately elevated plasma glucagon in both the non-fasted and fasted states, and impaired glucose-mediated glucagon suppression [166]. This is related to the loss of a distinct first phase of GSIS [167] and the negative paracrine impact of this on glucagon secretion. However, there is also evidence of a reduced responsiveness of alpha-cells to negative regulators of glucagon release in T2D, including insulin, serotonin, and somatostatin [18]. This may also be due to a reduced expression of SSTR2 receptors [168]. The increased glucagon release potentiates hepatic glucose production, further driving hyperglycemia and exacerbating beta-cell stress. Alpha-cell hyperplasia can also occur in humans and rodent models of T2D induced through obesity or prolonged hyperglycemia, although the more relevant measurement is likely to be the ratio of alpha-cell to beta-cell mass, especially in large islets, which strongly favours alpha-cells in T2D [169,170]. Hyperplasia of alpha-cells is also accompanied in rodent models by an increase in expression of PC1/3 and a relative switch in preproglucagon processing towards the secretion of GLP-1 [171,172]. This is likely to involve an increased inflammatory environment within the islets during T2D, since interleukin-6 (IL-6) increased both alpha-cell hyperplasia and GLP-1 release [173,174]. While the increase in circulating glucagon in T2D will amplify beta-cell stress through hyperglycemia, the local paracrine actions are likely to be beneficial; firstly, through a stimulatory action of glucagon on GSIS, and secondly, because GLP-1 can promote both beta-cell proliferation (at least in rodents), survival, and insulin expression and release [174,175]. 

An additional paracrine action of alpha-cell-derived GLP-1 in T2D could be to drive the generation of new beta-cells from resident progenitor cells, de-differentiated beta-cells or alpha-cells [176]. Both human and rodent beta-cells can de-differentiate when exposed to hyperglycemic stress to display a progenitor-like cell phenotype [177,178]. This is associated with a loss of expression of insulin and phenotypic beta-cell transcription factors such as PDX1 and MAFA, as well as GLUT2 [179]. A fuller phenotypic de-differentiation of beta-cells to become glucagon-expressing cells has been described in human T2D [180], but whether these represent fully functional alpha-cells is not known. Partial de-differentiation may be represented by an increased presence of bi-hormonal cells expressing both insulin and glucagon [181,182]. It is possible that de-differentiation of beta-cells may represent a pathway for beta-cell preservation in the face of prolonged glucotoxicity, provided that such cells are capable of redifferentiation to become functional beta-cells under less hyperglycemic conditions. Trans-differentiation of alpha- to beta-cells can occur in animal models of extreme beta-cell loss [183] and in isolated human islets from T2DM donors following transduction with PDX-1 or MAFA [184]. Functional ablation of the glucagon receptor in mice resulted in increased beta-cell mass involving trans-differentiation from alpha-cells [185]. GLP-1 action can facilitate alpha- to beta-cell trans-differentiation [186,187] and this potential is likely related to the many shared genes that are bivalently epigenetically marked and capable of reversal through histone modification [188]. 

Delta-cell mass and function are also altered in T2D, with animal models showing elevated somatostatin secretion at low glucose concentrations and a loss of amplified somatostatin release in the presence of glucose [189,190], with similar observations being reported for isolated human islets from subjects with T2D [97]. This will contribute to inappropriately elevated post-prandial glucagon release by the alpha-cell. It is not clear if this dysfunction is due to altered glucose sensitivity at the delta-cell or a decrease in paracrine secretion in the face of dysregulated insulin and UCN3 release. In the db/db rat model, T2D delta-cell mass initially increases with the appearance of hyperglycemia but is subsequently decreased as diabetes emerges [191]. A decreased delta-cell abundance has also been reported in non-human primates with pre-diabetes [192] and humans with T2D, with cells showing pathological morphology and hormone degranulation [193,194]. Results are divergent between species with respect to delta-cell mass in diabetes, as delta-cell hyperplasia was reported in adult Goto-Kakizaki diabetic rats associated with an increased expression of somatostatin [195]. Moreover, delta- to beta-cell trans-differentiation in adult diabetic mice has been reported, whereby the delta-cells exhibited an increased expression of the transcription factor Pax 4, which was associated with trans-differentiating cells [196]. 

Patients with T2D have a reduced number of islet epsilon cells, and fasted plasma levels of ghrelin are reduced, although it is not possible to discern how much of this circulatory ghrelin reduction is derived from islet production [197]. Isolated human islets from donors with T2D had a lower expression of ghrelin, although the glucose-dependent suppressive effect of exogenous ghrelin did not differ between donors with or without T2D. Conversely, the expression of the GHS-R was upregulated on beta-cells during diabetes [139], although the major paracrine action of ghrelin within the islet is considered to be on somatostatin release. Patients with T2D have high circulating levels of PP [198]. As PP can inhibit somatostatin production from human islets, this might represent a secondary contributor to the hyperglucagonemia of diabetes [133].

## 7. Islet Paracrine Disruption in T1D

The development of T1D has a strong genetic predisposition, but the timing of onset can be heavily dependent on environmental factors, including dietary components, stress, and viral infections. Genetic risk involves variability within the HLA system, with over 200 genes found on human chromosome 6, representing around half of the genetic risk as determined by genome-wide association studies [199]. HLA gene structures control the individual phenotype and affinity of T-cell receptors, affecting the functioning of the immune system and autoimmune tolerance. The HLA class II loci genes expressed in B lymphocytes, macrophages, dendritic cells, and the islets of Langerhans are particularly important for the recognition of islet protein autoantigens [200]. However, non-HLA gene polymorphisms also contribute to the risk of T1D, including the INS gene variable number of tandem repeats, PTPN22, CTLA4, and others [201,202,203,204]. A T1D susceptibility genotype results in reduced central and peripheral immune self-tolerance, leading to a progressive cascade of T-cell proliferation, activation, and insulitis. Viral triggers can precipitate the autoimmune destruction of beta-cells in individuals at genetic risk, including enteroviral infection by coxsackievirus B4, rotavirus and cytomegalovirus [205,206,207,208].

The major humoral autoantigens identified in the progression of T1D are insulin, glutamate decarboxylase, IA2, and the zinc transporter-8 [209,210,211,212]. Their combined presence in serum is a strong predictor of the presence and progression of T1D. Also, tetraspanin-7 has been detected in approximately 20–40% of T1D patients, with up to a 50% prevalence in diagnosed children [213]. Insulitis and CD8^+^/CD4^+^-T-cell-mediated cytotoxicity underlie beta-cell loss in human T1D. In rodent models of T1D, such as the autoimmune non-obese diabetic (NOD) mouse and BioBreeding (BB) rat, immune cell infiltration increases as the disease progresses until most islets display extensive infiltration. However, available histology banks of pancreatic tissue from human donors with T1D, such as the nPOD (network for Pancreatic Organ Donors) and Exeter Archival Diabetes Biobank, suggest that in human T1D, intense insulitis is largely absent [214]. There is evidence of exocrine involvement in the inflammatory pancreatic environment before the appearance of autoantibodies or clinical markers, indicating that the immune axis is disrupted early in the disease process [215].

Beta-cells are particularly susceptible to autoimmune cytotoxic damage due to the relatively high endoplasmic reticular and oxidative stress that exists under normal physiology as a consequence of the high metabolic activity required for continuous synthesis of large quantities of insulin and rapid exocytosis into the islet vascular bed [216]. This results in beta-cells being sensitive to even small changes in the islet inflammatory environment and the presence of cytotoxic CD8^+^ and CD4^+^ T-cells [217]. Inflammation causes the release of chemokines and cytokines from beta-cells that further amplify immune cell infiltration into the islets [218]. A response from beta-cells to a cascading inflammatory environment is to increase HLA class 1 expression, which amplifies the variety of immunogenic misfolded or incorrectly post-transcriptionally processed peptides, such as proinsulin, further increasing cytotoxic inflammatory cell presence [219]. As autoimmune insulitis progresses, cytokines released from the beta-cells themselves, such as IL-1beta and IL-6, can trigger autocrine/paracrine-induced apoptosis [220]. As beta-cells are lost, the oxidative stress experienced by remaining cells increases with the added demand for insulin release. As in T2D, some beta-cells may de-differentiate, with a loss of phenotypic gene expression such as MAFA and PDX1, improving their ability for survival [52], but ultimately the majority of beta-cells will succumb by the time of clinical diagnosis. Thus, the paracrine environment of the islet is progressively changed during T1D through increased paracrine availability of endogenous and exogenous cytotoxic cytokines. Inflammatory dysfunction results in a loss of inter-cellular coordination, whereby hub cells normally initiate waves of intracellular Ca^2+^ through gap junctions and pulsed insulin release. This disruption must also result in substantial alterations to the islet microvasculature as the intimate interfaces between beta-cells and the fenestrated endothelium are progressively lost.

In humans, the consensus thus far suggests that despite the proportion of alpha-cells remaining either the same [221,222] or being increased [223,224,225] within islets due to the loss of the beta-cells and a reduction in islet area, the overall alpha-cell mass remains similar in pancreata from donors with T1D when compared to non-diabetic donors. In contrast, Bonnet-Serrano and colleagues [226] showed alpha-cell mass in T1D to be significantly lower in a large-scale human study comprising pancreatic tissue from 75 patients with T1D and 66 normo-glycemic control donors from the publicly available nPOD tissue bank. The authors attributed this lower T1D alpha-cell mass to a reduced pancreatic weight in T1D when compared to the non-diabetic state. The discrepancy in the above findings could be attributed to differences in the sampling region, sampling methods, and quantification methods used. No difference in alpha-cell abundance was found between islets still containing beta-cells, where 33% also showed extensive insulitis, and islets devoid of beta-cells with little insulitis. However, Rahier et al. [223] reported an increased number of smaller islets and endocrine clusters in T1D, which contained a high proportion of alpha-cells. This was particularly apparent in pancreatic polypeptide-deficient areas of the pancreas, which also demonstrated the greatest extent of beta-cell loss.

In rodent models of T1D, such as major ablation of beta-cells with STZ, alpha-cell mass was reportedly increased [227], although it was unaltered in a repeated low-dose STZ model and in the progression of diabetes in the NOD mouse [228,229]. The latter two models involve insulitis, which may limit an increase in alpha-cell mass. In the RIP-B7.1 mouse model of experimental autoimmune diabetes, no difference in alpha-cell mass was reported after one week of diabetes compared to controls, but a subsequent increase in alpha-cell proliferation was found. As diabetes progressed, pancreatic glucagon content was significantly reduced in diabetic mice [221].

Despite the significant depletion of the beta-cells, beta-cell exocytosis and both first- and second-phase GSIS largely remain preserved in T1D when assessed from donor intact islets or pancreatic slices [230,231,232,233], but secrete less insulin when compared to non-diabetic islets (~1/60th of normal non-diabetic islets) due to their lower insulin content [234,235]. Moreover, human T1D islets also responded to amino acids, 3-isobutyl-1-methyl-xanthine (IBMX), and KCl-mediated depolarisation in a similar pattern to that of non-diabetic islets [230,231].

Unsurprisingly, as the beta-cells succumb to autoimmune-mediated destruction in T1D, the alpha-cells lose several beta-cell-derived paracrine inhibitory inputs, such as insulin, GABA, and serotonin. This inevitably results in a hypersecretion of glucagon in the presence of glucose, resulting in post-prandial hyperglucagonemia [233], exacerbating the diabetic hyperglycemia. As a result, individuals with T1D display similar levels of fasting glucagon to those without diabetes, despite having elevated fasting blood glucose [234,236]. Furthermore, carbohydrate-rich meals [233,235] and oral glucose tolerance tests [237] fail to suppress glucagon secretion in individuals with T1D when compared to normoglycemic individuals. Conversely, despite the preservation of the alpha-cells, individuals with T1D fail to secrete enough glucagon in the counterregulatory response to hypoglycemia when compared to non-diabetic individuals [236,238]. This was observed in some individuals with T1D who otherwise showed intact counter-regulatory responses to other counterregulatory hormones released during hypoglycaemia, such as growth hormone and cortisol [238]. The phenomenon is unlikely to be caused by a defect in the alpha-cell secretory machinery, as this would not be compatible with the increase in post-prandial glucagon levels in response to carbohydrates. Moreover, amino acids such as arginine and alanine still robustly stimulated glucagon secretion in individuals with T1D during insulin-infused hypoglycemic clamp studies [230,239,240].

A number of hypotheses have been put-forth to explain this counterregulatory defect in alpha-cell glucagon secretion in response to hypoglycaemia, which include: a loss of islet autonomic innervation; defects in neural glucose-sensing; alpha-cell insensitivity or irresponsiveness to paracrine signalling from beta-cells; an impairment in autocrine alpha-cell signalling or intrinsic defects from the alpha-cells themselves; as well as abnormally excessive somatostatin secretion from delta-cells [241]. Although limiting due to the sparse availability of tissue, recent studies using isolated T1D donor islets and pancreatic slices have shown alpha-cells to lack an appropriate increase in [Ca^2+^]_i_ and glucagon secretion in the presence of low glucose despite showing no difference in glucagon content when compared to non-diabetic donor islets/slices [242,243], suggesting that the mechanism(s) contributing to this glucagon secretory defect resides within the islets. This is supported by the observations that impaired counterregulatory glucagon secretion in response to hypoglycemia still occurs in humans with spinal cord transections [244] and in the denervated transplanted human pancreas [245]. Moreover, rodent models of beta-cell ablation, such as treatment of rats with a high-dose STZ, also show an attenuated glucagon response to insulin-induced hypoglycaemia in vivo, suggesting the importance of beta-cell presence for normal counter-regulatory alpha-cell glucagon secretion [246,247].

Recently, impaired autocrine glutamate-AMPA/kainate receptor signalling from the alpha-cells has been proposed by Panzar and colleagues [242] to play a critical role in the loss of counter-regulatory glucagon secretion from T1D islets in response to hypoglycemia. This was demonstrated to be caused by a chronic desensitization of the alpha-cell AMPA/kainate receptors. Intriguingly, by inhibiting AMPA/kainate receptor desensitisation by exposing T1D donor pancreatic slices to cyclothiazide (a positive allosteric modulator of AMPA/kainate receptors approved by the FDA for edema treatment), the authors showed that alpha-cell [Ca^2+^]_i_ and glucagon secretion during hypoglycaemia could be significantly improved [242]. Moreover, transcriptomic studies comparing isolated alpha-cells from T1D versus non-diabetic donors revealed a significant down-regulation in transcripts required for sufficient cAMP signalling; potassium and sodium, as well as Ca^2+^ channel formation and function; and glucose metabolism during T1D [225]. In addition, this study showed that alpha-cells from T1D donors contained a reduced expression of multiple nuclear regulators, including ARX, MAFB, NKX2.2 and RFX6; with the latter transcription factor (RFX6) being an identified upstream regulator of both MAFB and ARX. which in mature human and rodent beta-cells controls the expression of P/Q and L-type voltage-gated Ca^2+^ channels (CACNA1A, CACNA1C, and CACNA1D), as well as the K_ATP_ channel subunit, sulfonylurea receptor 1 (ABCC8) [225]. Intriguingly, several reports have identified alpha-cells from T1D donors as expressing NKX6.1, a transcription factor normally specifically expressed in beta-cells [225,248]. This raises the possibility that in T1D alpha-cells may also be re-programmed to a partial beta-cell phenotype, as reported in rodent models of induced diabetes [183].

Delta-cell mass is largely unaltered during T1D, although delta-cell presence is proportionately increased as beta-cells are lost [249]. It has been suggested that increased paracrine-mediated alpha-cell inhibition by excessive basal somatostatin secretion from the delta-cells in response to low glucose may be responsible for the aberrant glucagon response to hypoglycemia in T1D. Plasma somatostatin levels are reportedly elevated in alloxan-induced diabetic dogs [250] and STZ-induced diabetic rats [189] when compared to normo-glycaemic control animals. However, the short half-life of somatostatin of under a minute in the circulation and the major circulatory source being from the gut and central nervous system, with only 5–10% being derived from the islet delta-cells, mean that plasma levels are a poor reflection of pancreatic somatostatin synthesis [251]. Yue and colleagues [246] demonstrated using high-dose STZ-injected diabetic rats subjected to a hypoglycemic clamp that the attenuated glucagon response to insulin-induced hypoglycemia could be significantly improved by the simultaneous application of a SSTR2 antagonist (SSTR2a), confirming an elevation in basal somatostatin release and paracrine feedback inhibition on the alpha-cells. This observation was subsequently confirmed in vivo and in pancreatic slices in vitro using the autoimmune diabetic BB rat, supporting an elevated presence of delta-cell-derived somatostatin within the diabetic intra-islet milieu [252]. SSTR2 is widely expressed in the body, and therefore the therapeutic application of SSTR2a requires clinical safety testing. SSTR2a’s are currently used as a diagnostic tool for positron emission tomography imaging for the identification of neuroendocrine tumours, where they are reported to be well-tolerated [253]. A SSTR2a, ZT-01, produced by Zucara Therapeutics is currently in phase II clinical trials for preventing nocturnal hypoglycemia in individuals with T1D after successfully completing a phase I clinical study whereby 90% of T1D subjects undergoing a hypoglycemic clamp demonstrated an improved glucagon response in the presence of ZT-01 [254,255]. SSTR2a’s have also been shown to restore alpha-cell counterregulatory glucagon secretion in rodents subjected to antecedent hypoglycemia, suggesting that paracrine delta-cell somatostatin feedback inhibition on the alpha-cells increases with each recent bout of hypoglycaemia [247,256]. The use of SSTR2a’s to restore glucagon counter regulation in diabetes has been recently reviewed by Hoffman and colleagues [257].

Reports on the paracrine functions of other islet endocrine cell types in T1D are sparse. However, in a rat model of STZ-induced loss of beta-cell mass, the epsilon-derived peptides, ghrelin and obestatin, were both protective against beta-cell loss and resulted in an upregulation of Ins1 and Pdx1 expression in the remaining beta-cells, allowing a limited regeneration of beta-cell mass [115,258].

## 8. Pregnancy and Gestational Diabetes

While a limited increase in beta-cell mass can occur through the generation of new cells in human T2D through hypertrophy, the capacity for enhanced proliferation in adults is limited [259]. The exception to this is pregnancy, where the increased maternal insulin resistance necessary to provide glucose flux across the placenta for the growing fetus necessitates a substantial adaptive increase in beta-cell mass and function [260,261]. During human pregnancy, maternal insulin resistance increases across gestation under the influence of placentally derived variant growth hormone and other placental hormones [262], necessitating an increase in maternal insulin-secreting capacity in order to maintain euglycemia. To avoid beta-cell stress, this is accomplished by a substantial increase in beta-cell mass under the trophic influence of human placental lactogen and prolactin [263]. Human studies are few given the need for post-mortem tissue from women during pregnancy, but two such studies exist. Van Assche et al. [264] reported that beta-cell fractional area was 2.4-fold greater in pregnant women compared to non-pregnant controls. A second study by Butler et al. [265] reported a 1.4-fold increase in beta-cell fractional area, although the mean age of gestation of the subjects was lower. Further, this study found that the major source of new beta-cells was likely to be islet neogenesis, giving rise to many small islets. Metabolic adaptations during pregnancy also involve lowering the glucose threshold for GSIS as well as beta-cell hypertrophy and greater insulin biosynthesis [266,267].

Rodents also demonstrate an adaptive increase in beta-cell mass during pregnancy, although the fold increase is greater than in humans and involves the recruitment of previously proliferatively-quiescent beta-cells into the replicative cycle [33,267]. The increase in beta-cell mass is preceded by capillary endothelial cell proliferation, with these endothelial cells releasing soluble factors in vitro that have a greater proliferative action on co-cultured beta-cells than from islet endothelial cells harvested outside of pregnancy [146]. Endothelium-secreted products capable of promoting beta-cell proliferation included HGF. The increase in beta-cell proliferation is also enabled through the prolactin receptor in response to external stimulation by placental lactogen and prolactin signaling [268], but these lactogens also modify the paracrine secretome of the beta-cell, involving increased secretion of serotonin, resulting in autocrine/paracrine activation of 5-HT2B receptors on beta-cells to activate cell cycle progression [269]. Experimental loss of islet serotonin does not, however, completely block the beta-cell proliferative response to pregnancy [63]. A generation of new beta-cells from resident progenitor cells may also contribute to the increased beta-cell mass in the mother, as we have demonstrated for immature cells lacking GLUT2 but expressing low levels of insulin, which increased early in pregnancy in the maternal islets [33]. Trans-differentiation of the alpha-to beta-cell phenotype does not seem to be a contributor to increasing beta-cell mass during normal rodent pregnancy [270,271]. Co-release of UCN3 together with insulin from beta-cells outside of pregnancy provides a paracrine feedback mechanism to inhibit both somatostatin and glucagon release in order to fine-tune subsequent GSIS. During mouse pregnancy, the placenta also releases UCN2 and 3 into the maternal circulation from mid-gestation, which might supplement the actions of the islet-derived peptides in controlling glucose homeostasis [272].

The adaptive islet changes during pregnancy also involve non-beta-cell populations. Alpha-cell mass increases during a healthy mouse pregnancy, involving both hypertrophy and proliferation, which are maximal in late gestation, with relative hyperglucagonemia compared to the non-pregnant state [271,273,274,275]. The pancreatic content of both glucagon and GLP-1 increases during mouse pregnancy [274]. As with beta-cells, remodeling of alpha-cell mass has also been attributed to the actions of the gestational hormones placental lactogen, prolactin and estradiol [273]. Targeted depletion of alpha-cells during pregnancy resulted in impaired glucose tolerance, whereas glucose tolerance was improved in non-pregnant mice [274]. Treatment with a GLP-1 receptor agonist improved insulin release and glycemic control in alpha-cell null mice. It is therefore possible that alpha-cell-derived GLP-1 is an important driver of beta-cell function in the adaptive changes of pregnancy. Protein restriction during pregnancy in rodents impaired the adaptive increase in beta-cell mass and resulted in a reduced content of intra-islet GLP-1, resulting in impaired glucose tolerance [273,275]. The proportional presence of delta-cells did not change during mouse pregnancy, suggesting that delta-cell mass must increase in absolute terms across gestation relative to the increase in beta-cell mass, and the proportion of delta cells increased still further in a diabetes model [276]. Following pregnancy, both beta- and alpha-cell mass return to pre-pregnancy levels through targeted apoptosis in rodents [266,277], and presumably in humans as well.

Failure to adequately undergo the adaptive changes in beta-cell mass and function during pregnancy can precipitate GDM. Pre-gestational obesity or excessive weight gain during pregnancy represent risk factors for GDM, and the added insulin resistance associated with obesity can overwhelm the adaptive potential of the endocrine pancreas [278]. Expansion of beta-cell mass can be impaired in a diabetogenic environment, ultimately resulting in glucose intolerance [273]. Deficient beta-cell function in the first trimester is a risk factor for the development of GDM [279,280]. In rodent models of GDM, glucose intolerance in late pregnancy correlated with reduced beta-cell replication in mid-gestation, a smaller mean islet size and beta-cell mass at term, and impaired GSIS [273]. The increase in alpha-cell mass was also impaired in a mouse model of GDM [270,273].

## 9. Overview

The islets of Langerhans have evolved in mammals to become anatomically dispersed mini-organs, self-contained and self-regulating in their autocrine and paracrine control of the release of the major systemic hormones regulating glucose metabolism, insulin and glucagon. Functional heterogeneity exists between individual and regional populations of endocrine cells within a single islet and between individual islets within a pancreas, with each islet having the ability to self-regulate intra-islet blood flow to meet overall metabolic demand. These anatomically distributed yet coordinated metabolic regulators are acutely responsive to minute-by-minute changes in circulating glucose and amino acids, and CNS input through innervation provides longer-term awareness of metabolic status—famine or feast. Whilst not the focus of this review, pancreatic blood flow also allows for connectivity of humoral signaling between the exocrine and endocrine components of the pancreas [281], which is likely to contribute to the development of insulitis and T1D. The finely tuned self-regulating machine that is the islet is precariously balanced due to the high metabolic rate required by beta-cells to synthesize and secrete insulin and can succumb to oxidative and endoplasmic reticular stress in the face of metabolic challenges posed either by obesity or an autoimmune environment.

However, expanding our understanding of islet paracrine interactions offers new strategies for preventing or reversing islet dysfunction or loss, and the detrimental impact of altered hormonal balance during diabetes. To date, little attention has been given to the minority islet endocrine cell populations. The intra-islet role of the delta-cells and the impact of dysregulated somatostatin secretion on the defective counter-regulatory alpha-cell glucagon response to hypoglycaemia in T1D are excellent examples of a potentially novel tool in diabetes control. While beta-cell plasticity in terms of proliferation is limited in adults outside of pregnancy, there is evidence that during diabetes, adaptive mechanisms exist to maximize beta-cell survival and function through a relative switch in post-translational processing of preproglucagon in alpha-cells towards GLP-1 secretion. This also has the potential to promote phenotypic trans-differentiation of alpha-cells into beta-cells. The ability of beta-cells to enter dormancy during metabolic and/or cytotoxic stress may allow them to survive for reactivation in a future protective environment. Such an environment may also include the presence of ghrelin and obestatin. Similarly, islet adaptations to the metabolic stress of pregnancy have evolved to respond to placentally derived lactogenic hormones exerting trophic actions on beta-cells through the prolactin receptor. This mechanism utilizes an increased secretion of paracrine-acting serotonin from the beta-cells, which might be manipulated to preserve beta-cell mass. However, as with any complex machine, when one variable is changed, the effect reverberates through every control element of the structure. This is likely to also be the case with the islets of Langerhans.

## Figures and Tables

**Figure 1 ijms-25-04070-f001:**
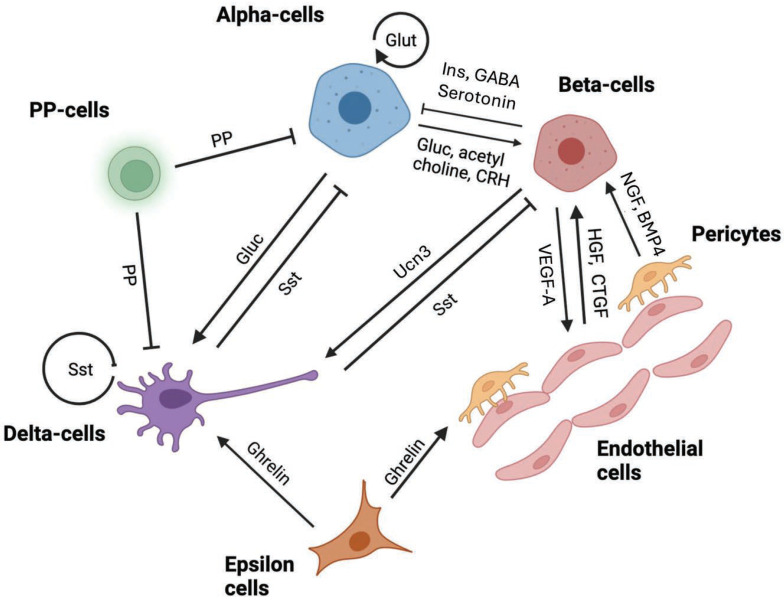
Major paracrine interactions within islets of Langerhans involving beta-cells, alpha-cells, delta-cells, epsilon cells, pancreatic polypeptide cells, endothelial cells and pericytes contributing to the regulation of hormonal release or cell proliferation/survival. Insulin—Ins, glucagon—Gluc, somatostatin—sst, pancreatic polypeptide—PP, corticotrophin releasing hormone—CRH, urocortin-3-UCN3, glutamate—Glut, vascular endothelial growth factor-A—VEGF-A, hepatocyte growth factor—HGF, connective tissue growth factor—CTGF, nerve growth factor—NGF, and bone morphometric protein-4—BMP4. Arrows indicate predominantly positive effects (→) or negative actions (⊣).

**Table 1 ijms-25-04070-t001:** Changes in paracrine islet interactions reported to occur during T1D, T2D and GDM.

	T2D	T1D	GDM
Beta-cells	Reduction in BCM due to gluco-lipotoxicity. Deposition of amyloid. Senescence-associated secretory phenotype. Loss of coordinated GSIS response. Reduced UCN3 secretion. GSH-R abundance increased.	Reduction in BCM due to insulitis and cytokine toxicity. Dedifferentiation of remaining beta-cells. Loss of hub cells and coordinated insulin release.	Insufficient adaptive increase in BCM in response to lactogenic hormones. Reduced islet presence of beta-cell progenitors. Reduction in GSIS due to gluco-lipotoxicity. Reduced autocrine actions of serotonin on insulin release.
Alpha-cells	Alpha-cell hyperplasia. Increased glucagon secretion and delayed glucose-mediated glucagon suppression. Increased GLP-1 secretion. Alpha- to beta-cell transdifferentiation.	Little change in alpha-cell mass. Post-prandial hyper-glucagonemia following loss of negative effects of beta-cell-derived insulin, GABA and serotonin. Reduction in glucose suppression of glucagon release. Impaired glutamate-AMPA/kainate receptor signalling.	Insufficient adaptive increase in alpha-cell mass in response to lactogenic hormones.
Delta-cells	Decreased delta-cell abundance. Loss UCN3-stimulated somatostatin release. Elevated somatostatin release at low glucose. Delta- to beta-cell transdifferentiation.	Unaltered delta-cell mass. Increased basal somatostatin release. Increased inhibition of alpha-cell glucagon release.	Increase in delta-cell mass during pregnancy may be impaired.
Epsilon/PP-cells	Reduced epsilon cell number and ghrelin release. Increased circulating PP.	Protective effect of ghrelin and obestatin on beta-cell survival and insulin release.	No information.
Islet vasculature	Capillary thickening and increased vessel density. Loss of pericytes.	Loss of capillary/beta-cell interfaces.	Insufficient adaptive increase in capillary density.

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
