# Peer review of "The Importance of Intra-Islet Communication in the Function and Plasticity of the Islets of Langerhans during Health and Diabetes"

_ijms, 2024, doi:10.3390/ijms25074070_

Round 1

Reviewer 1 Report

Comments and Suggestions for Authors

The manuscript "The importance of intra-islet communication in the function of the islets of Langerhans during health and diabetes" focuses on the intercellular communication that regulates the function of cells in the pancreatic islets under normal conditions and its dysregulation during different types of diabetes. The review addresses the topic clearly and deeply, allowing for the presentation and integration of novel information and its correlation with different types of diabetes. The manuscript is correctly written and contains clear language.

Minor Observations: Line 52 - Replace the term "innovated" with "innervated". Figure 1 - The image appears distorted. Additionally, correct the term "Gluc" which is included as part of beta cell secretion and affects (inhibits) the functioning of alpha cells. Line 78 - Remove comma before reference [4]; and correct "10K" to "100". Line 80 - Correct "5K" to "50". Line 287 - Replace "amino acids" with "fatty acids". Line 338 - Add the name of the receptor and place the initials in parentheses. "glucosphyngolipid receptor (GSL-R)" Line 391 - Reference [150] does not correspond to the information. The reference mentions IAPP but not pancreatic fat deposition. Reference PMID: 34671102 may be useful. Line 520 - Remove one "[" symbol from reference [[217]. Line 521 - Replace "Il-1" with "IL-1". Line 692 - Replace the term "β-cells" with "beta cells". Line 695 - Remove hyphen in the term "beta-cells" and replace with "beta cells".

Minor Suggestions:

Section "Islet paracrine disruption in T2D": It is suggested that the authors include a paragraph mentioning the possible role of the imbalance in the proportion of immature and mature beta cells that could alter renewal of the beta cell mass and/or function. Some useful reviews about this topic include PMID: 34667280; PMID: 33586491; PMID: 27585958; and PMID: 31419404.

Additionally, authors may include a section focused on the intra-islet signaling impairment produced by beta-cell senescence associated secretory phenotype (SASP.)

Overview Section: It is suggested that the authors modify the sentence "The ability of beta cells to enter senescence during metabolic and/or cytotoxic stress may allow them to survive for reactivation in a future protective environment" and consider removing the part mentioning "reactivation of the cell cycle"; it should be noted that senescence is defined as the process by which a cell exits the cell cycle "permanently and irreversibly" (even with stimulation) PMID: 37229454.

Author Response

Reviewer 1

Minor Observations: Line 52 - Replace the term "innovated" with "innervated". Figure 1 - The image appears distorted. Additionally, correct the term "Gluc" which is included as part of beta cell secretion and affects (inhibits) the functioning of alpha cells. Line 78 - Remove comma before reference [4]; and correct "10K" to "100". Line 80 - Correct "5K" to "50". Line 287 - Replace "amino acids" with "fatty acids". Line 338 - Add the name of the receptor and place the initials in parentheses. "glucosphyngolipid receptor (GSL-R)" Line 391 - Reference [150] does not correspond to the information. The reference mentions IAPP but not pancreatic fat deposition. Reference PMID: 34671102 may be useful. Line 520 - Remove one "[" symbol from reference [[217]. Line 521 - Replace "Il-1" with "IL-1". Line 692 - Replace the term "β-cells" with "beta cells". Line 695 - Remove hyphen in the term "beta-cells" and replace with "beta cells".

We thank the reviewer for picking up these textual errors. All have been corrected. With reference to GSL-R. In that section we meant to refer to the GHS-R receptor only, not GSL-R. This has been corrected. The misattribution of reference 150 has been corrected by addition of the citation that the reviewer recommended. We looked at again at Figure 1 but cannot see any distortion. We will submit again in case the file was corrupted .

Minor Suggestions:

Section "Islet paracrine disruption in T2D": It is suggested that the authors include a paragraph mentioning the possible role of the imbalance in the proportion of immature and mature beta cells that could alter renewal of the beta cell mass and/or function. Some useful reviews about this topic include PMID: 34667280; PMID: 33586491; PMID: 27585958; and PMID: 31419404.

All of these recommended papers have been checked. While the substantial heterogeneity in beta-cell sub-groups includes FLTP , ST8S1A1, PSA-CAM or GLUT2 expression, bi-hormonal presence, and other markers; there does not seem to be a consensus as to whether these represent different stages of maturity of beta-cells or simply functional heterogeneity. Similarly, there does not seem to be reproducibility as to how the relative populations change in T2D. We are very receptive to the idea that there might be a change in the balance of immature to mature beta-cells in T2D we do not think the evidence stands up yet to support this.  

Additionally, authors may include a section focused on the intra-islet signaling impairment produced by beta-cell senescence associated secretory phenotype (SASP.)

We thank the reviewer for this good suggestion. A paragraph around SASP and T2D has been added on page 21.

Overview Section: It is suggested that the authors modify the sentence "The ability of beta cells to enter senescence during metabolic and/or cytotoxic stress may allow them to survive for reactivation in a future protective environment" and consider removing the part mentioning "reactivation of the cell cycle"; it should be noted that senescence is defined as the process by which a cell exits the cell cycle "permanently and irreversibly" (even with stimulation) PMID: 37229454.

Corrected as suggested.

Reviewer 2 Report

Comments and Suggestions for Authors

The review manuscript  from Thomas G. Hill and David J. Hill, “The importance of intra-islet communication in the function and plasticity of the islets of Langerhans during health and diabetes”, gives a comprehensive overview of the many years of research on various aspects of  paracrine interactions within islets of Langerhans contributing to the regulation of hormonal release or cell proliferation/survival during normal and abnormal glycemic control . The review is scientifically sound, well-planned, and written in an organized manner. The aim of the review article is well discussed and supported with previously published data. The limitations of current research in this area and proposing avenues for future investigation are also addressed. The references are appropriate and sufficient. However, it would be beneficial to include the final table summarizing the different mechanisms of  islet paracrine disruption in T1D, T2D and gestational diabetes.

Author Response

Reviewer 2

However, it would be beneficial to include the final table summarizing the different mechanisms of  islet paracrine disruption in T1D, T2D and gestational diabetes.

Good suggestion. A Table has been added.